# Microstructure and Mechanical Properties of Hot-Extruded Mg–Zn–Ga–(Y) Biodegradable Alloys

**DOI:** 10.3390/ma15196849

**Published:** 2022-10-02

**Authors:** Viacheslav Bazhenov, Anna Li, Stanislav Tavolzhanskii, Andrey Bazlov, Natalia Tabachkova, Andrey Koltygin, Alexander Komissarov, Kwang Seon Shin

**Affiliations:** 1Casting Department, National University of Science and Technology “MISiS”, Leninskiy pr. 4, 119049 Moscow, Russia; 2Laboratory of Hybrid Nanostructured Materials, National University of Science and Technology “MISiS”, Leninskiy pr. 4, 119049 Moscow, Russia; 3Laboratory of Advanced Green Materials, National University of Science and Technology “MISiS”, Leninskiy pr. 4, 119049 Moscow, Russia; 4Department of Materials Science of Semiconductors and Dielectrics, National University of Science and Technology “MISiS”, Leninskiy pr. 4, 119049 Moscow, Russia; 5Fianit Laboratory, Laser Materials and Technology Research Center at GPI, Prokhorov General Physics Institute RAS, Vavilov st. 38, 119991 Moscow, Russia; 6Laboratory of Medical Bioresorption and Bioresistance, Moscow State University of Medicine and Dentistry, Delegatskaya 20/1, 127473 Moscow, Russia; 7Magnesium Technology Innovation Center, Department of Materials Science and Engineering, Seoul National University, 1 Gwanak-ro, Gwanak-gu, Seoul 08826, Korea

**Keywords:** biomaterials, gallium, hot extrusion, magnesium, mechanical properties, microstructure

## Abstract

Magnesium alloys are attractive candidates for use as temporary fixation devices in osteosynthesis because they have a density and Young’s modulus similar to those of cortical bone. One of the main requirements for biodegradable implants is its substitution by tissues during the healing process. In this article, the Mg–Zn–Ga–(Y) alloys were investigated that potentially can increase the bone growth rate by release of Ga ions during the degradation process. Previously, the effectiveness of Ga ions on bone tissue regeneration has been proved by clinical tests. This work is the first systematic study on the microstructure and mechanical properties of Mg–Zn–Y alloys containing Ga as an additional major alloying element prepared by the hot-extrusion process. The microstructure and phase composition of the Mg–Zn–Ga–(Y) alloys in as-cast, heat-treated, and extruded conditions were analyzed. In addition, it was shown that the use of hot extrusion produces Mg–Zn–Ga–(Y) alloys with favorable mechanical properties. The tensile yield strength, ultimate tensile strength, and elongation at fracture of the MgZn4Ga4 alloy extruded at 150 °C were 256 MPa, 343 MPa, and 14.2%, respectively. Overall, MgZn4Ga4 alloy is a perspective for applications in implants for osteosynthesis with improved bone regeneration ability.

## 1. Introduction

Permanent bone fixation implants are the gold standard in osteosynthesis and are used in healthcare systems in many countries. Titanium alloys have been used for bone fixation implants. However, the titanium systems have disadvantages including temperature sensitivity, tactile sensation of implants, possible growth restrictions, hampering of imaging and radiotherapy, presence of titanium particles in surrounding tissue, and potential mutagenicity [1]. These disadvantages result in symptomatic removal in up to 40% of cases [1]. At present, temporary biodegradable implants are gaining popularity, that gradually dissolve as the healing process progresses [2]. This approach helps to minimize implant-induced inflammation and reduces healthcare costs by eliminating secondary surgery for implant removal [3]. Mg alloys are attractive candidates for the fabrication of temporary fixation devices for osteosynthesis. They have good biocompatibility, sufficiently high mechanical strength, and an acceptable biodegradation rate [4,5,6]. Furthermore, unlike Ti implants for permanent bone fixation, Mg alloys have a similar density and Young’s modulus to cortical bone [7,8].

At present, the commercial NOVAMag^®^ and MAGNEZIX^®^ fixation screws produced by Botiss biomaterials GmbH (Berlin, Germany) and Syntellix AG (Hannover, Germany) are used in orthopedic practice and showed equal performance in tissue regeneration with Ti alloys [9,10]. One of the main capabilities of biodegradable implants is the same rates of implant degradation and bone tissue growth for substituting the voids with new bone tissue. Several studies revealed Mg has a positive impact on the bone regeneration [2], but additional efforts can increase the Mg implants’ applications. For example, previously, a study on animals showed that Ga-contained hydroxyapatite coating on Gription™ implants (West Chester, PA, USA) increased the bone growth rate by two times [11]. However, the coating used for biodegradable implants will improve osteogenesis only before it dissolves. The main objective of this work is proposing the new alloy system with the addition of a component that can improve the bone tissue growth process. The most appropriate component in this method is gallium. Gallium is known as a bone resorption inhibitor that increases the Ca and P content in the developed bone [12,13]. Thus, it effectively treats osteoporosis [14], hypercalcemia [15,16,17], Paget’s disease [18,19], and multiple myeloma [20]. In addition, Ga has an anti-osteoclastic effect that reduces osteoclastic resorption, differentiation, and formation without negatively affecting the osteoblast’s viability and proliferation [21,22,23].

When the release of Ga ions occurs during the degradation of Mg–Zn–Ga–(Y) alloys, and local Ga delivery to tissues takes place. Thus, the Mg–Zn–Ga–(Y) alloys in bone fixation implants can improve the bone healing process, which is the benefit in comparison with other conventional magnesium alloys. The design and introduction of a new implant into medical practice require deep knowledge of the mechanical, corrosion, and biological performance of the new material [24].

The effect of Zn addition on Mg alloys properties is well known [25,26], but the effect of Ga is currently under investigation. Liu et al. showed that Ga decreased the c/a proportion of magnesium solid solution (α-Mg), leading to the activation of additional slip planes and improved plastic deformation of Mg alloys [27]. Furthermore, the addition of Ga can reduce the critical strain for dynamic recrystallization (DRX) in Mg alloys, which is associated with a reduced stacking fault energy, increased twinning density during deformation, and an increase in the number of DRX nucleation sites [28,29]. Gallium also acted as an effective grain refiner [30]. It is known that the work hardening ability of Mg–Ga alloys is higher than that of pure Mg due to a uniform fine equiaxed microstructure with a low dislocation density [31]. Thus, Ga addition to Mg is expected to lead to excellent mechanical properties [31]. Furthermore, the addition of Zn to Mg–Ga alloys reduces the activation energy barrier for nucleation of the Mg_5_Ga_2_ phase, resulting in more and finer precipitates [32].

Previously, Mg–4Zn–4Ga (wt.%) alloys with small additions of Ca, Y, and Nd were investigated after equal channel angular pressing (ECAP). It was found that these alloys had high strength (up to 300 MPa) and a low corrosion rate of ~0.2 mm/year in Hanks’ solution [33]. The addition of Y does not decrease the corrosion rate of Mg–Zn–Ga alloys compared to that of Ca and Nd [33]. Further, Y addition contributes to enhanced protective properties and a higher alloy ignition temperature [34]. Therefore, the Mg–Zn–Ga alloy with Y addition needs to be investigated further.

Despite our previous study on Mg–Zn–Ga–(Y) alloys after ECAP processing, the potential of other deformation processing techniques such as hot extrusion on alloy properties has not yet been thoroughly examined [33]. The hot extrusion process has various advantages compared with ECAP: less limitations in size and shape of the billet, easy control of microstructure, low processing cost, high yield, etc. Due to this, the hot extrusion is better fit to high-volume manufacturing. Thus, the aim of the study was to investigate the effects of the chemical composition and extrusion temperature on the microstructure and mechanical properties of Mg–Zn–Ga–(Y) alloys. This work is the first systematic study on the properties of Mg–Zn–(Y) alloys containing Ga as an additional major alloying element prepared by the hot-extrusion process to evaluate their potential for application in orthopedic implants.

The extrusion processing temperature window is usually determined by the possibilities of equipment used (possible lowest extrusion temperature) and limit of hot cracking (possible highest extrusion temperature) [35]. For Mg–Zn–Ga–(Y) alloys, the solidus temperature is close to 300 °C, which was shown previously [33] and confirmed via CALPHAD calculation and DSC in this work. Due to this, the upper limit of extrusion temperature 250 °C was chosen. The lower limit of 150 °C was chosen in accordance with the maximal pressure of the used press.

## 2. Materials and Methods

### 2.1. Alloy Preparation and Hot Extrusion

The scheme of preparing samples for investigation is presented in Figure 1. High-purity bulk metals, including Mg (99.98 wt.% purity; SOMZ, Solikamsk, Russia), Zn (99.995 wt.%; UGMK, Verkhnaya Pyshma, Russia), Ga (99.9999 wt.%; Girmet Ltd., Moscow, Russia), and Mg–20Y (wt.%) master alloy (Metagran, Moscow, Russia) were used as the starting materials for the preparation of alloys. The melts were prepared using a resistance furnace with a steel crucible. For melt protection from ignition, an Ar + 2 vol.% SF_6_ protective atmosphere was used. The details of the melting procedure can be found elsewhere [36]. Cylindrical ingots with a diameter of 60 mm and a length of 200 mm were cast into an aluminum permanent mold preheated to 150 °C. Five alloys with different Zn and Ga contents were prepared, as listed in Table 1. In addition, an alloy with the addition of Y was prepared. The chemical compositions of the alloys were analyzed using energy-dispersive X-ray spectroscopy (EDS) on the metallographic sections with 0.1 wt.% accuracy. The three areas with size 1 × 1 mm^2^ were analyzed for each specimen.

To dissolve the eutectic phases and homogenize the ingot composition, a solid solution heat treatment (HT) at 300 °C for 15 h + 400 °C for 30 h was applied [33]. First, the ingots were machined into cylindrical billets with a height of 145 mm and diameter of 50 mm. Next, hot extrusion of the alloys was performed on a 300 ton vertical hydraulic press PS-300A7 (Gidrosfera, Moscow, Russia) using the direct extrusion method at a ram speed of 1 mm/s and an extrusion ratio of 6.25 (Figure 1). Finally, cylindrical extruded bars with a diameter of 20 mm and length of ~1 m were obtained. Before the hot extrusion process, the die was preheated to 200–250 °C. Extrusion was performed at billet temperatures of 150, 200, and 250 °C to evaluate the effect of this temperature of the final alloy properties.

### 2.2. Microstructural Observations, Phase Composition, and Thermal Analysis

The phase composition and phase transition temperatures of the Mg–Zn–Ga–(Y) alloys were calculated according to the calculation of phase diagram (CALPHAD) method using FactSage software (GTT-Technologies, Aachen, Germany). Furthermore, the Scheil–Gulliver solidification of the alloys was calculated [37]. The thermodynamic database FTlite (GTT-Technologies, Aachen, Germany) was used.

The alloy microstructure and elemental content of the phases were investigated using scanning electron microscopy (SEM; Vega SBH3, Tescan, Brno, Czech Republic) with an EDS detector (Oxford, UK) and transmission electron microscopy (TEM; JEM-2100, JEOL, Tokyo, Japan). The accelerating voltage used for TEM investigations was 200 kV. The samples for TEM were prepared using ion-beam etching, which was performed using a Precision Ion Polishing System PIPS II (Gatan, Pleasanton, CA, USA) at a voltage of 3 keV. The phase volume fractions were determined by calculating the area of phases in the SEM image using Tescan software (Tescan, Brno, Czech Republic). The grain size of the extruded samples was determined using the linear intercept method, with the assistance of SEM and an Axio Observer D1m (Carl Zeiss, Oberkochen, Germany) optical microscope (OM). To reveal the grains, the metallographic cross-sections were etched for 5 s using an etchant (11 g picric acid, 11 mL acetic acid, and 100 mL ethanol).

The solidus temperatures of the alloys were measured using differential scanning calorimetry (DSC; Labsys Setaram, Caluire, France) under an Ar gas flow at a heating rate of 20 °C/min in Al_2_O_3_ crucibles.

The alloy phase compositions were examined using bulk cylindrical specimens with X-ray diffractometry (XRD) with a D8 ADVANCE diffractometer (Bruker, Billerica, MA, USA) under monochromatic Cu Kα radiation.

### 2.3. Mechanical Properties

The mechanical properties of the alloys were investigated using a 5569 universal testing machine (Instron, Norwood, MA, USA) equipped with an advanced video extensometer. The tolerance of testing machine was less than 0.5%. Tensile tests were performed using two types of specimens: standard cylindrical specimens obtained by extruded bar lathe machining and small flat plate tensile test specimens produced by wire cutting the hot-extruded bars [38]. The compression tests were performed using small cuboid specimens (3 mm × 3 mm × 6 mm). The small flat plate and cuboid specimens were cut along and perpendicular to the extrusion direction (ED). Three standard cylindrical specimens and six to eight flat plates and cuboid specimens were tested for each alloy and extrusion temperature.

## 3. Results and Discussion

### 3.1. Quality of Extruded Bars

The quality of the extruded bars is shown in Figure 2a. The bars were divided into three groups by defectiveness: bars with large radial cracks visible by the naked eye (Figure 2b), bars with surface cracks with a depth of 1−2 mm (Figure 2c), and bars of good quality with no visible cracks on the surface (Figure 2d). Increasing the alloying element content and extrusion temperature decreased the quality of the bars. During extrusion, the deformation energy is converted into heat, thereby increasing the billet temperature above the extrusion temperature. If the billet temperature is close to the solidus temperature, crack formation is possible. Therefore, the residual eutectic in the high-alloyed alloys that melts during extrusion at 250 °C results in crack formation (Figure 2b).

### 3.2. Microstructure and Phase Composition

Table 2 shows liquidus and solidus temperatures, fraction of phases at room temperature (RT) and temperatures when secondary phases start to precipitate calculated with FactSage software. The increase in alloying elements’ content leads to a decrease in equilibrium solidus and liquidus temperature. In accordance with calculation results, the RT microstructure of investigated alloys must comprise α-Mg, Mg_5_Ga_2_, and Mg_12_Zn_13_ (otherwise known as MgZn). When Y is added, the I phase (Mg_3_Zn_6_Y) is precipitated. Furthermore, for high alloyed alloys, the precipitation temperature of Mg_5_Ga_2_ and Mg_12_Zn_13_ phases and its fraction at RT are higher than for their low alloyed counterparts. This means that precipitation hardening must be greater for MgZn4Ga4, MgZn4Ga4Y0.5, and MgZn6.5Ga2 alloys.

The calculation of the solidification pathway using the Scheil–Gulliver solidification model shows that in all of investigated alloys, the ternary eutectic transition L→α-Mg + Mg_12_Zn_13_ + Mg_5_Ga_2_ occurs at 299 °C. Therefore, the two-stage solution heat treatment was used for alloys, where the first stage at 300 °C is for dissolution of non-equilibrium eutectic.

The microstructures of the alloys in the as-cast condition are shown in Figure 3. All alloys contained α-Mg dendrites with clear visible micro-segregation and eutectic intermetallic phases. The microstructure and EDS maps in Figure 4 show the Zn, Ga, and Y distributions for the MgZn4Ga4 and MgZn4Ga4Y0.5 alloys. The microstructure of MgZn4Ga4 showed two eutectic phases with a composition close to the Mg_51_Zn_20_ (otherwise known as Mg_7_Zn_3_) and Mg_5_Ga_2_ phases, as determined by EDS point analysis. According to the phase diagram, when the temperature is lower than 325 °C, Mg_51_Zn_20_ does not exist, and the Mg_12_Zn_13_ phase precipitates [39]. However, in magnesium alloys with Zn addition, the Mg_51_Zn_20_ intermetallic phase could be found in the microstructure of the as-cast alloys. Further, it will be seen that after solution heat treatment and hot extrusion, the Mg_12_Zn_13_ phase is present in microstructures rather than Mg_51_Zn_20_, in accordance with the Mg–Zn phase diagram.

The addition of Y to MgZn4Ga4 leads to formation of the Ga–Y eutectic phase. The obtained results were similar to those observed previously [33,39,40,41,42], where the Ga–Y phase observed. Furthermore, this phase is not rich in Zn and consists mostly of Y and Ga. It differs from the FactSage calculation results, where I phase (Mg_3_Zn_6_Y) rich in Zn formation is predicted.

The alloy microstructures after HT are also shown in Figure 3. In comparison with as-cast microstructure, the dissolution of intermetallic phases and changes in their morphology to spherical structures were observed. The fractions of Mg_12_Zn_13_, Mg_5_Ga_2_, and Ga–Y intermetallic phases in Mg–Zn–Ga–(Y) alloys in the as-cast condition and after HT are present in Figure 5a. After HT, a residual intermetallic phase was observed for high-alloyed alloys (MgZn6.5Ga2, MgZn4Ga4, and MgZn4Ga4Y0.5). For low-alloyed MgZn4Ga2 and MgZn2Ga2 alloys, a single-phase α-Mg microstructure was observed. In the microstructure of the alloys after HT, Mg_12_Zn_13_, Mg_5_Ga_2_, and Ga–Y intermetallic phases are present. These phases act as cathodes for α-Mg during the corrosion process [33]. Therefore, high volume fractions of such phases in the alloy are undesirable from the corrosion standpoint. In opposite, the intermetallic phases can act as recrystallization centers according to the particle-simulated nucleation mechanism during subsequent deformation and lead to a decrease in grain size [43].

The contents of Zn and Ga in the α-Mg of the as-cast and heat-treated Mg–Zn–Ga–(Y) alloys obtained using EDS analysis are shown in Figure 5b. The Zn and Ga contents in the α-Mg after HT were close to the expected nominal contents of these elements in the alloy. This is in agreement with the near-full dissolution of the intermetallic phases in the alloys (Figure 5a). In contrast, the Y content in α-Mg for the MgZn4Ga4Y0.5 alloy before and after HT was around 0.05 wt.% and is not shown in Figure 5b. This is in accordance with our previous observations and indicates that the Ga–Y phase is not dissolved during HT [33].

The DSC results for Mg–Zn–Ga–(Y) alloys in the as-cast and HT conditions are shown in Figure 6. The solidus temperature for all investigated alloys obtained from as-cast samples was 316 °C, which is associated with the ternary eutectic transition L→α-Mg + Mg_51_Zn_20_ + Mg_5_Ga_2_ [33]. This temperature is close to the ternary eutectic transition temperature of 299 °C obtained via FactSage calculation. In the post-HT alloys, the endothermic effect connected with the eutectic melting was not observed. During HT, the intermetallic phases formed due to eutectic transition dissolved, and the solidus temperature of the alloys increased. However, in the alloy microstructure produced under HT conditions (Figure 5a), complete intermetallic phase dissolution was observed for low-alloyed MgZn4Ga2 and MgZn2Ga2 alloys only. The high-alloyed alloys after HT still contained intermetallic phases in their microstructures. The absence of an endothermic peak at 316 °C in the DSC curve for these alloys after HT is associated with a low fraction of intermetallic phases in their microstructure leading to a low thermal effect of eutectic melting.

The microstructures along the direction parallel to the ED of the Mg–Zn–Ga–(Y) alloys extruded at 150 and 200 °C are shown in Figure 3. The alloys extruded at 250 °C had similar microstructures (not shown here). In the extruded alloys, the intermetallic phases, which did not fully dissolve during HT, were elongated towards the ED and were significantly fragmented. MgZn2Ga2 and MgZn4Ga2, which had a single-phase microstructure after HT, were the alloys with no large intermetallic phases observed. The insets in Figure 3 (pink boxes) show the microstructures of the extruded alloys at high magnification. Small precipitates of secondary phases were formed during extrusion via a stress-induced precipitation mechanism in the α-Mg matrix.

TEM images of the hot-extruded Mg–Zn–Ga–(Y) alloys are shown in Figure 7. A large amount of near-spherical precipitates, 100–200 nm in diameter, were observed in the MgZn4Ga4 alloy extruded at 200 °C (Figure 7a). The precipitates were located both inside the grains and at the grain boundaries. According to the EDS results, the compositions of the precipitates are close to the Mg_5_Ga_2_ and Mg_12_Zn_13_ phases, which form agglomerates, and Figure 7a also shows these agglomerates at higher magnification.

In the MgZn2Ga2 alloy extruded at 150 °C, only Mg–Ga precipitates were found, but their compositions and sizes were different (Figure 7b). Most of the precipitates had a size of 20–50 nm and their composition was close to that of Mg_5_Ga_2_. However, large precipitates (up to 1 μm) and compositions close to Mg_2_Ga_5_ or MgGa were also observed, but their volume fraction was low. The Mg_5_Ga_2_ precipitates also formed agglomerates. Thus, the large fraction of small precipitates must provide high mechanical properties.

The insets in Figure 3 in the blue box show the microstructure of the alloys after etching. The grain structure was close to a fine DRX one, but a small fraction of coarse non-DRX grains was also observed. The average grain sizes of the Mg–Zn–Ga–(Y) alloys extruded at different temperatures are shown in Figure 8. The grain sizes of the MgZn4Ga4 alloys formed at extrusion temperatures of 150 or 200 °C were not measured because the samples could not be effectively etched due to high fraction of intermetallic phases in its structure. However, it can be concluded that the finer grains were observed at lower extrusion temperatures, and for the MgZn2Ga2 alloy, it is more clearly seen.

Increasing the Zn content from 2 to 6.5 wt.% resulted in a three-fold decrease in the grain size. In contrast, Ga had little effect on the grain size of the hot-extruded alloys. The Mg_5_Ga_2_ and Mg_12_Zn_13_ phases were dynamically precipitated during extrusion and can act as pinning obstacles in the growth of DRX grains via the Zener drag effect [44,45]. However, Mg_12_Zn_13_ precipitates are more effective for forming a fine grain structure. Previously, it was shown that MgZn4Ga4 after ECAP at 310 °C has a bimodal structure that comprises large grains surrounded by small grains [33]. In contrast with ECAP, extrusion of the alloy promoted unimodal microstructure formation.

Figure 9 shows the XRD spectra of the Mg–Zn–Ga–(Y) alloys after extrusion at 150 °C. The extrusion temperature had little influence on the phase composition of the investigated alloys, and the XRD spectra of the alloys after hot extrusion at 200 or 250 °C, were the same as those at 150 °C and are not shown. In all alloys, the α-Mg, Mg_5_Ga_2_, and Mg_12_Zn_13_ phases were observed, as confirmed by the TEM EDS results of the extruded alloys (Figure 7). Further, the XRD peaks of the Mg_2_Ga_5_ and MgGa phase were not observed for the extruded alloys because of their low fraction compared to the Mg_5_Ga_2_ phase. The alloy with added Y in extruded state contained a Ga–Y phase (the peaks corresponds to both Ga_6_Y and Ga_3_Y_5_) that confirmed the EDS results of as-cast alloy and opposite to CALPHAD calculation results via FactSage. This can be connected with FTlite thermodynamic database limits.

### 3.3. Mechanical Properties

Typical engineering tensile stress–strain curves obtained for large standard cylindrical specimens of the Mg–Zn–Ga–(Y) alloys after hot extrusion at 150 or 200 °C are shown in Figure 10, while the corresponding tensile properties are shown in Figure 11a–c. From the tensile stress–strain curves, the slope of the elastic region was the same for all investigated alloys.

For high-alloyed alloys (MgZn4Ga4, MgZn4Ga4Y0.5, MgZn6.5Ga2), the tensile yield strength (TYS; Figure 11a) increased when the extrusion temperature increased from 150 to 200 °C. We observed that the alloys extruded at 150 °C had smaller grain sizes, and the higher TYS values could be due to a higher quantity or effectiveness of the Mg_12_Zn_13_ and Mg_5_Ga_2_ precipitates formed during extrusion. For low-alloyed alloys (MgZn4Ga2 and MgZn2Ga2), no difference in the TYS between alloys extruded at 150 and 200 °C was observed because of a small amount of precipitates and their minimal effect on TYS. Increasing the extrusion temperature to 250 °C led to a decrease in the TYS, which was attributed to the large grain size (Figure 8). It can be seen from Figure 11b,c that the extrusion temperature has little effect on the ultimate tensile strength (UTS) and elongation at fracture (El). The exceptions were MgZn4Ga4 and MgZn6.5Ga2 alloys, which have low El (<10%), but show the highest TYS. The addition of Y promoted a decrease in TYS. As shown previously, Y was not dissolved in α-Mg during HT, and the Ga–Y phase precipitates remained in the microstructure and decreased TYS.

An increase in the Zn and Ga contents significantly affected the tensile properties of the alloys. The high-alloyed alloys showed higher TYS and UTS than their low-alloyed counterparts, but the El curves were very similar for the investigated alloys. For example, the maximal and minimal mechanical properties were observed for MgZn4Ga4 and MgZn2Ga2 alloys, respectively. These results are in good agreement with the content of alloying elements and the grain size of the alloys (Figure 8), where the fine grain structure corresponded to better tensile properties.

Hot-extruded materials have anisotropic properties in directions parallel and perpendicular to the ED. It is known that loads can be applied to bone implants in different directions, and the complex stress–strain distributions can also change during implant degradation. Therefore, the material properties that are applied to implants should be investigated in different directions. The tensile properties of the extruded Mg–Zn–Ga–(Y) alloys as a function of extrusion temperature in directions parallel and perpendicular to the ED were obtained using small flat-plate specimens, as shown in Figure 11d–f. Similar TYS and UTS values were obtained from the large standard cylindrical specimens and the small flat-plate ones cut parallel to the ED. However, El was lower when small flat-plate specimens are used, which could be due to the higher surface area to volume ratio for small specimens and the higher density of surface stress concentrators. The influence of the alloy composition on the anisotropy of the mechanical properties was negligible. As expected, for all alloys and extrusion temperatures, the tensile properties obtained for the specimens cut perpendicular to the ED were inferior to those cut along the ED. The minimal anisotropy of the tensile properties was observed for alloys extruded at the highest temperature (250 °C), while the maximal anisotropy of UTS and El was observed for alloys extruded at 200 °C. This anisotropy in the mechanical properties could be related to the deformation texture and non-DRX grains in the alloy structure. The maximal difference for the TYS, UTS, and El values obtained parallel and perpendicular to the ED were 90 MPa, 160 MPa, and 6%, respectively. Hence, the anisotropy of the mechanical properties should be considered when using Mg–Zn–Ga–(Y) alloys for bone implants.

Implants are generally subjected to alternating loads, and it is essential to investigate their compressive properties. The compressive yield strength (CYS) and compressive strength (CS) were investigated as a function of the extrusion temperature (Figure 11g,h, respectively) using small cuboid specimens of the extruded Mg–Zn–Ga–(Y) alloys with the long axis cut parallel or perpendicular to the ED direction. With increasing extrusion temperature from 150 to 200 °C, the change in CYS was insignificant, but a further increase in temperature led to a decrease in CYS. The CS decreased for most alloys with increasing extrusion temperature. The maximum and minimum values of both CYS and CS were observed for MgZn4Ga4 and MgZn2Ga2 alloys, respectively. Overall, the effects of the extrusion temperature and alloy composition on the tensile and compression properties were similar. The maximal difference of CYS measured in the directions parallel and perpendicular to ED was 50 MPa, which was lower than a difference obtained for the tensile test results. As expected, the CS of the alloys was higher than that of the UTS for all alloys. The yield asymmetry (CYS/TYS) was in the range of 0.74–0.99 for the investigated alloys. Typically, the deformation texture and grain size affect the yield asymmetry [46]. We observed that the yield asymmetry was slightly lower for alloys extruded at 150 °C than those extruded at 200 °C. With increasing extrusion temperature from 150 to 200 °C, TYS increased, but CYS remained the same. The lowest yield asymmetry was observed for MgZn4Ga4 (~0.95), that probably connected with the maximal fraction of small recrystallized grains of alloy and lower texture in ED after deformation processing [35].

In previous work, it was shown that the solution heat treatment of the Mg–Zn–Ga–(Y) alloys induced a high El (up to 15.2%) and produced a low fraction of intermetallic phases that provided high corrosion resistance [36]. However, aging had a low effect on the alloy strength and still has a low value. Therefore, the deformation processing is the only way to make the use of these alloys for biodegradable implants in osteosynthesis possible.

The YS and El of Mg–Zn–Ga magnesium alloys after extrusion are higher than its binary Mg–Zn and Mg–Ga counterparts [47,48]. This indicates that addition of both Ga and Zn has a positive effect on magnesium alloy’s strength. It was shown previously that the YS, UTS, and El of the MgZn4Ga4 alloy after ECAP at 310 °C were 165 MPa, 300 MPa, and 22%, respectively [33]. Based on the results of this work, this alloy, after extrusion at 150 °C, showed higher strength (YS = 256 MPa and UTS = 343 MPa) and lower El (14%). The enhancement in the mechanical properties after hot extrusion occurred due to grain refinement, which produces a significant effect on the properties of Mg alloys because of their high Hall–Petch strengthening coefficient (~300 MPa·μm^1/2^) [49]. The difference in the obtained properties is attributed to the lower deformation processing temperature and lower grain size for the hot extrusion process in comparison with ECAP processing.

This study was the first attempt to investigate the influence of hot extrusion on the microstructure and mechanical properties of newly developed Mg–Zn–Ga–(Y) alloys. For better understanding of the mechanical behavior of the mentioned alloys, extensive texture analysis and Visco-Plastic Self-Consistent (VPSC) simulations are needed. Furthermore, the implants’ work in the human body and the mechanical properties in the body fluids corrosive environment such as stress corrosion cracking and corrosion fatigue must be known for Mg–Zn–Ga–(Y) alloys in order to use them as biodegradable materials.

## 4. Conclusions

Five Mg–Zn–Ga–(Y) alloys with different Zn and Ga contents were prepared and subjected to hot extrusion. When appropriate extrusion temperatures (150 and 200 °C) were used, no defects were observed on the bars of low-alloyed alloys, in contrast with radial cracks observed on high-alloyed ones. The as-cast alloys contained α-Mg and two eutectic phases (Mg_5_Ga_2_ and Mg_51_Zn_20_). HT resulted in the near-complete disappearance of intermetallic phases and α-Mg with Zn and Ga contents close to those of the bulk alloy. Micro-alloying with Y promoted the formation of Ga–Y phase, which was retained after HT. The hot extrusion of the alloys resulted in the elongation and fragmentation of residual intermetallic phases and the formation of precipitates (Mg_5_Ga_2_ and Mg_12_Zn_13_). Increasing the Zn content in the alloys resulted in better grain refinement during deformation processing, but Ga did not affect the grain size.

Increasing the alloying-element content led to higher strength of the alloys. For instance, when Zn and Ga content in alloy extruded at 200 °C increased from 2 to 4 wt.%, the TYS and UTS increased by 115 and 71 MPa, respectively, but El decreased from 13.9 to 7.6%. However, the extrusion temperature had little effect on their mechanical properties. As expected for hot extrusion-processed alloys, the mechanical properties of the alloys were better along the ED than in the direction perpendicular to it. The TYS, UTS, and El values obtained perpendicular to the ED were decreased by 90 MPa, 160 MPa, and 6%, respectively, in comparison with the values obtained along the ED. Adding Y to Mg–Zn–Ga alloys had little effect on their properties after hot extrusion.

The best combination of mechanical properties is obtained for MgZn4Ga4 alloy extruded at 150 °C, for which the TYS, UTS, and El were 256 MPa, 343 MPa, and 14.2%, respectively, and compressive properties CYS and CS were 251 and 587 MPa. These values were on par with or better than those of several Mg alloys considered as candidate materials for bone replacement implants [33,47].

Further research will focus on in vitro corrosion behavior and biocompatibility of Mg–Zn–Ga–(Y) alloys to choose the best composition. After that, the coating must be developed for reduction in biodegradation rate and gas formation that provide better bone integration. Finally, in vivo testing on animals to show the increased bone growth rate and improved healing process in comparison with other magnesium alloys will be conducted.

## Figures and Tables

**Figure 1 materials-15-06849-f001:**
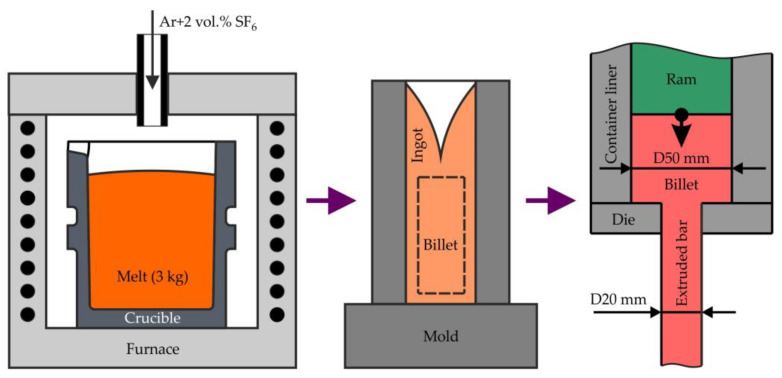
The scheme of extruded bars processing from casting to hot extrusion.

**Figure 2 materials-15-06849-f002:**
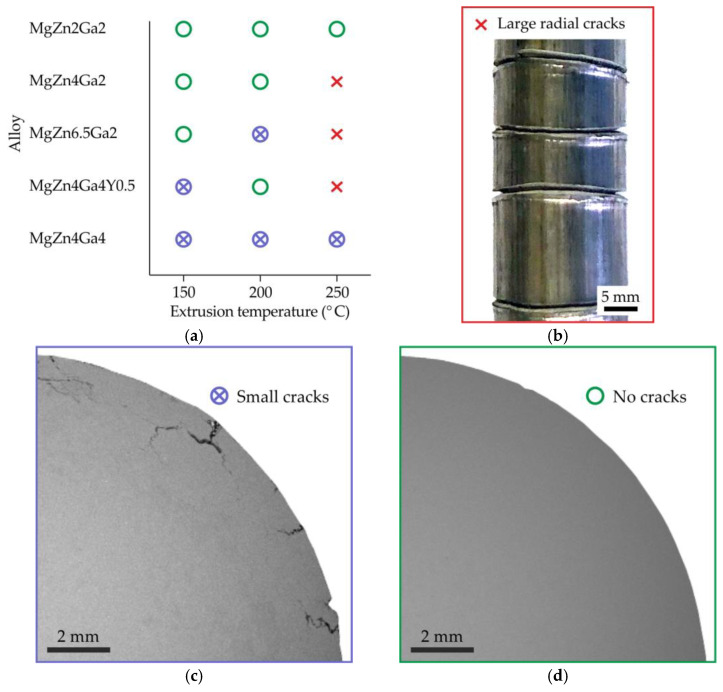
(**a**) Graph defining the quality of the Mg–Zn–Ga–(Y) alloy bars as a function of alloy composition and extrusion temperature. The quality examples are shown in: (**b**) bar with large radial cracks (e.g., MgZn4Ga2 alloy extruded at 250 °C), (**c**) bar with small (~1−2 mm) surface cracks (e.g., MgZn4Ga4 alloy extruded at 150 °C), and (**d**) bar without cracks (e.g., MgZn2Ga2 alloy extruded at 150 °C).

**Figure 3 materials-15-06849-f003:**
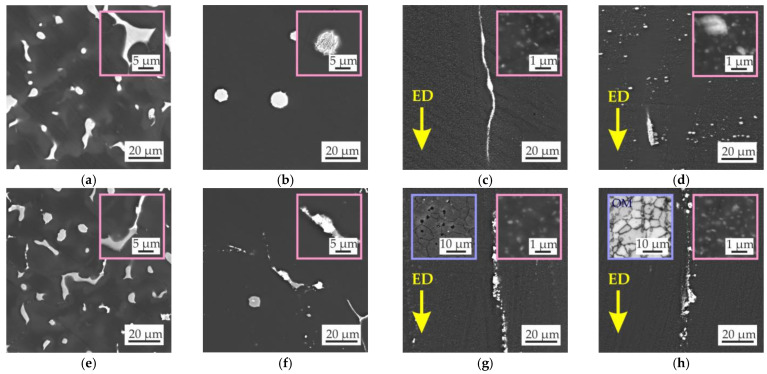
Microstructure of the (**a**–**d**) MgZn4Ga4, (**e**–**h**) MgZn4Ga4Y0.5, (**i**–**l**) MgZn6.5Ga2, (**m**–**p**) MgZn4Ga2, and (**q**–**t**) MgZn2Ga2 alloys in the (**a**,**e**,**i**,**m**,**q**) as-cast condition, (**b**,**f**,**j**,**n**,**r**) after HT for 15 h at 300 °C + 30 h at 400 °C, and after hot extrusion at (**c**,**g**,**k**,**o**,**s**) 150 or (**d**,**h**,**l**,**p**,**t**) 200 °C. The insets with pink boxes show higher-magnification images, while those in blue boxes show the microstructure after etching (only for extruded alloys).

**Figure 4 materials-15-06849-f004:**
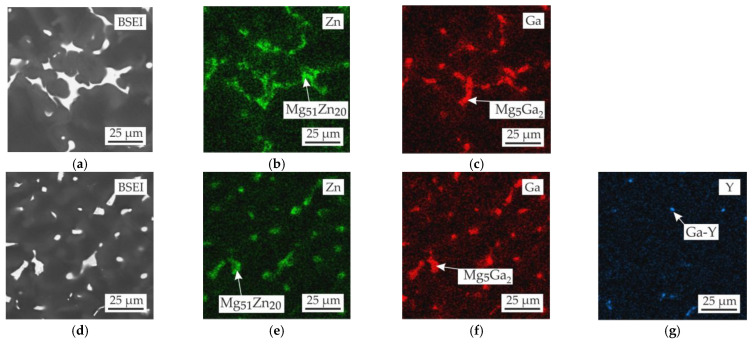
Microstructure and EDS maps of the (**a**–**c**) MgZn4Ga4 and (**d**–**g**) MgZn4Ga4Y0.5 alloys in the as-cast condition. The EDS maps showing the distribution of (**b**,**e**) Zn, (**c**,**f**) Ga, and (**g**) Y.

**Figure 5 materials-15-06849-f005:**
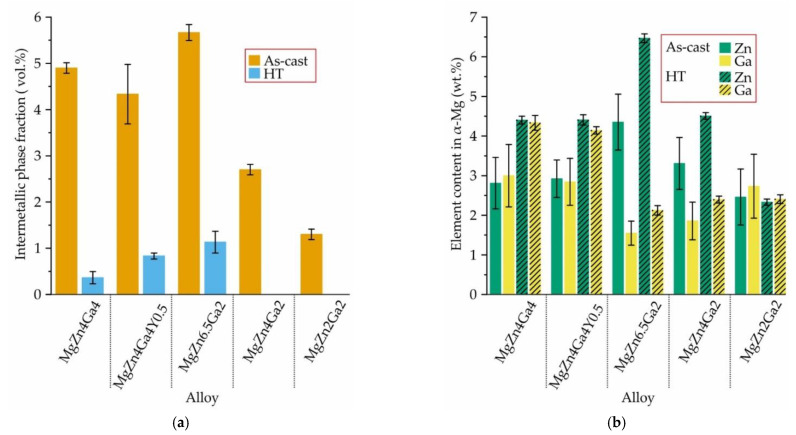
(**a**) Total fraction of intermetallic phases Mg_51_Zn_20_ (or Mg_12_Zn_13_ after HT), Mg_5_Ga_2_, and Ga–Y, and (**b**) content of Zn and Ga in the α-Mg solid solution for Mg–Zn–Ga–(Y) alloys in as-cast and HT conditions.

**Figure 6 materials-15-06849-f006:**
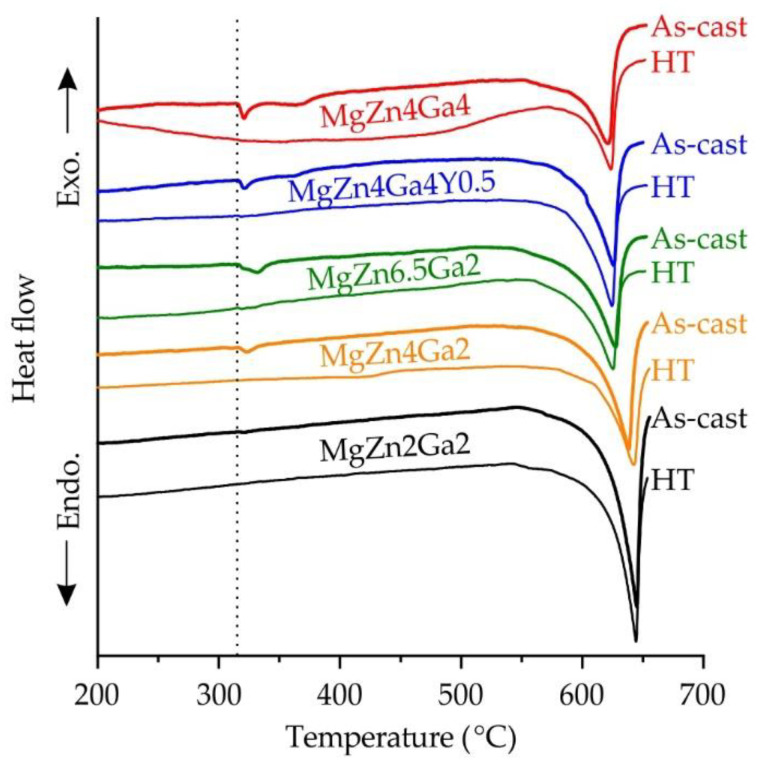
DSC heating curves for Mg–Zn–Ga–(Y) alloys in the as-cast condition and after HT.

**Figure 7 materials-15-06849-f007:**
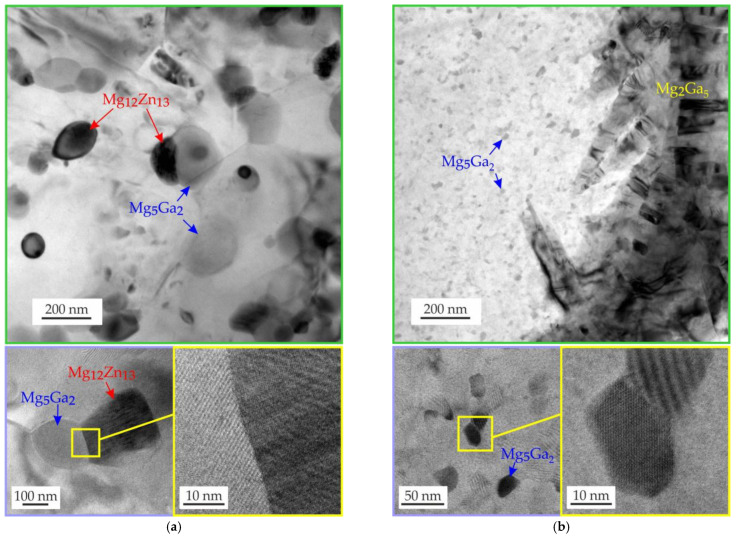
TEM images of (**a**) MgZn4Ga4 alloy extruded at 200 °C and (**b**) MgZn2Ga2 alloy extruded at 150 °C.

**Figure 8 materials-15-06849-f008:**
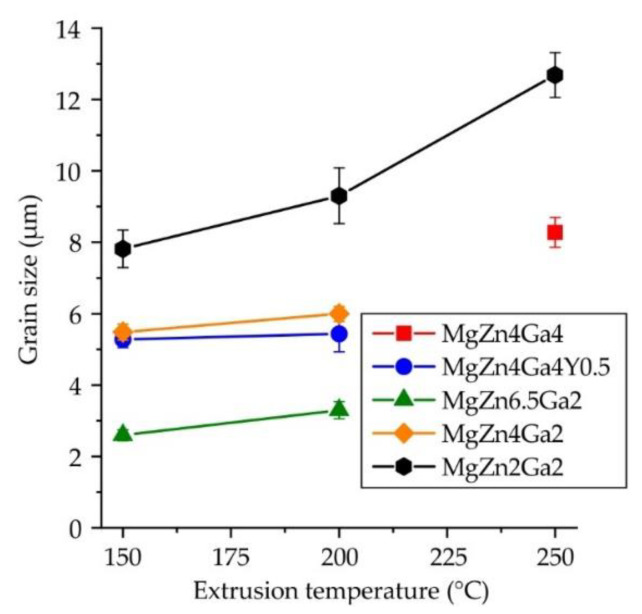
Average grain size vs. extrusion temperature of the Mg–Zn–Ga–(Y) alloys.

**Figure 9 materials-15-06849-f009:**
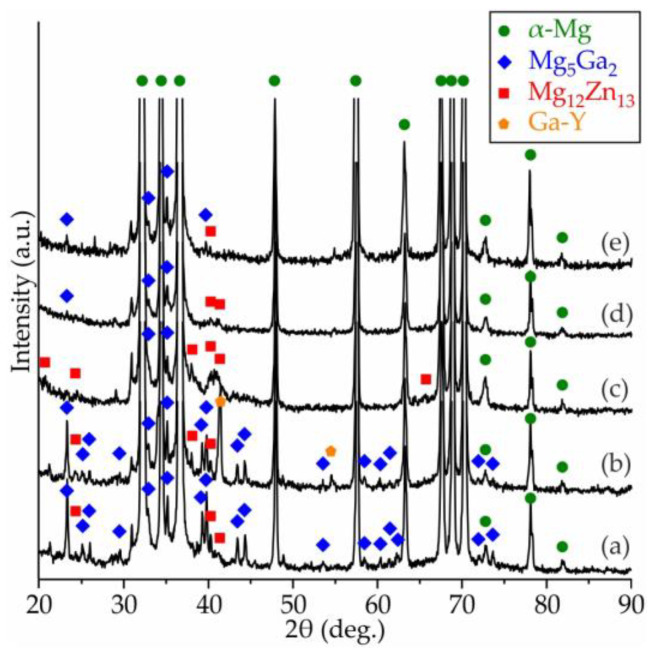
XRD patterns of (**a**) MgZn4Ga4, (**b**) MgZn4Ga4Y0.5, (**c**) MgZn6.5Ga2, (**d**) MgZn4Ga2, (**e**) MgZn2Ga2 after hot extrusion at 150 °C.

**Figure 10 materials-15-06849-f010:**
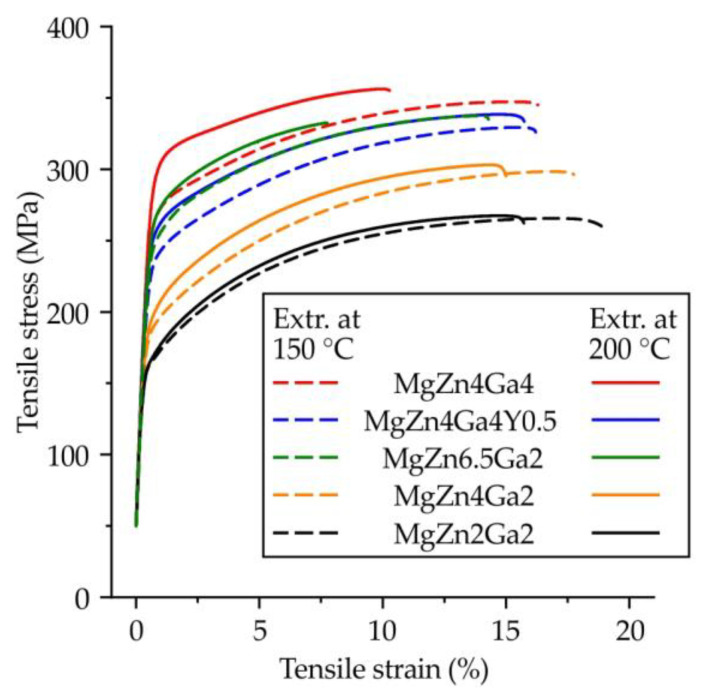
Engineering tensile stress–strain curves for the Mg–Zn–Ga–(Y) alloys extruded at 200 or 150 °C.

**Figure 11 materials-15-06849-f011:**
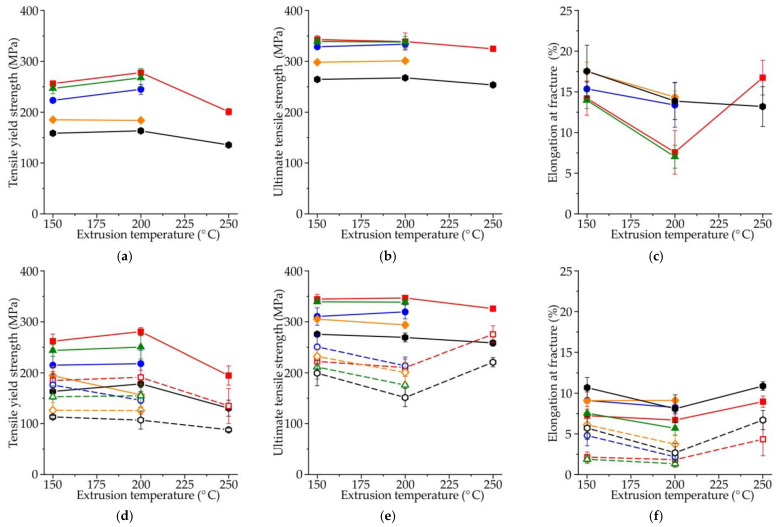
Mechanical properties as a function of the extrusion temperature of the Mg–Zn–Ga–(Y) alloys evaluated using (**a**–**c**) large standard cylindrical tensile-test specimens, (**d**–**f**) small flat-plate tensile-test specimens, or (**g**,**h**) small compression-test specimens cut along the directions parallel and perpendicular to the ED. (**a**,**d**) TYS; (**b**,**e**) UTS; (**c**,**f**) El; (**g**) CYS; and (**h**) CS; (**i**) legend.

**Table 1 materials-15-06849-t001:** Elemental compositions of the prepared alloys.

Alloy	Element Content (wt.%)			
	Mg	Zn	Ga	Y
MgZn4Ga4	Bal.	4.2	4.1	-
MgZn4Ga4Y0.5	Bal.	4.2	4.1	0.4
MgZn6.5Ga2	Bal.	6.5	2.0	-
MgZn4Ga2	Bal.	4.2	2.2	-
MgZn2Ga2	Bal.	2.3	2.3	-

**Table 2 materials-15-06849-t002:** The Mg−Zn−Ga alloys phase composition and phase transition temperatures calculated via FactSage software.

Alloy	T_liq.Sch_ (°C)	T_sol.eq_ (°C)	M_eut.Sch_ (wt.%)	Eq. Phase Amount at RT (wt.%)			Eq. Precipit. Start. Temp. (°C)		
				Mg_5_Ga_2_	Mg_12_Zn_13_	I	Mg_5_Ga_2_	Mg_12_Zn_13_	I
MgZn4Ga4	624	370.5	9.82	7.5	5.41	-	266	258	-
MgZn4Ga4Y0.5	623	412	6.58	7.5	3.05	2.49	264	189	434
MgZn6.5Ga2	623	334	8.02	3.57	8.5	-	182	321	-
MgZn4Ga2	630	421	8.70	3.94	5.41	-	188	250	-
MgZn2Ga2	636	497	5.05	4.12	2.85	-	189.5	174	-

## Data Availability

Not applicable.

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
