# Peer review of "Microstructure and Mechanical Properties of Hot-Extruded Mg–Zn–Ga–(Y) Biodegradable Alloys"

_materials, 2022, doi:10.3390/ma15196849_

Round 1
Reviewer 1 Report
1. The current title's uppercase and lowercase should be changed following MDPI format.
2. Emails from all authors are formatted using MDPI and are written in black without italics.
3. The abstract section should include quantitative results.
4. Please give a “take-home” message as the conclusion of your abstract.
5. Put the keywords in a new order based on alphabetical order.
6. Based on the MDPI format, all of the keywords must be written in lowercase.
7. It is encouraged not used abbreviations in the keywords section.
8. The novel of the present study is not clear. Many published literature has been widely studied in the past related to magnesium based biodegradable alloys. Further explanation in the introduction section in advance is mandatory.
9. Previous research has to be explained in the introduction section, including their work, novelty, and limits, to illustrate the research gaps that will be filled in the current study.
10. Why the present study perform experimental testing? Why not combined with/only computational simulation and/or clinical study?
11. Since the present study only perform experimental testing, the potential of computational simulation study via finite element method is needed to discuss since bring some advantages such as faster results and do not need sophisticate tools. The introduction and/or discussion part of an article should contain this crucial topic, according to the authors. In addition, to reinforce this explanation, the MDPI-recommended reference should be cite as follows: Ammarullah, M. I.; Afif, I. Y.; Maula, M. I.; Winarni, T. I.; Tauviqirrahman, M.; Jamari, J. Tresca Stress Evaluation of Metal-on-UHMWPE Total Hip Arthroplasty during Peak Loading from Normal Walking Activity. Mater. Today Proc. 2022, 63, S143–6. https://doi.org/10.1016/j.matpr.2022.02.055
12. In order to improve the reader's understanding of the materials and methods section simpler, the authors could provide a figures that clarify the workflow of the current study rather than only the predominant text as it currently appears.
13. Additional information about tools used, such as the maker, country, and specification, should be included.
14. The error and tolerance of the experimental tools used in this investigation are important aspects that have to be mentioned in the manuscript. It might be valuable for further research by other scholars because of the different results.
15. Outcomes must be compared to similar past research.
16. Before moving on to the conclusion section, the present study's limitation must be added at end of the discussion section.
17. In the conclusion, please explain the further research.
18. The reference should be enriched with literature from the last five years. Literature published by MDPI is strongly recommended.
19. Due to grammatical and linguistic style issues, the authors should proofread the manuscript. For this issue, the authors would utilize the MDPI English editing service.
20. Please ensure that the writers followed the MDPI format correctly, modify the current form, and recheck in addition to any other problems that have been identified.
Author Response
- The current title's uppercase and lowercase should be changed following MDPI format.
Answer: It was fixed.
- Emails from all authors are formatted using MDPI and are written in black without italics.
Answer: It was fixed.
- The abstract section should include quantitative results.
Answer: The abstract was rewritten.
- Please give a “take-home” message as the conclusion of your abstract.
Answer: The abstract was rewritten.
- Put the keywords in a new order based on alphabetical order.
Answer: It was fixed.
- Based on the MDPI format, all of the keywords must be written in lowercase.
Answer: It was fixed.
- It is encouraged not used abbreviations in the keywords section.
Answer: It was fixed.
- The novel of the present study is not clear. Many published literature has been widely studied in the past related to magnesium based biodegradable alloys. Further explanation in the introduction section in advance is mandatory.
Answer: The introduction was rewritten.
- Previous research has to be explained in the introduction section, including their work, novelty, and limits, to illustrate the research gaps that will be filled in the current study.
Answer: The introduction was rewritten.
- Why the present study perform experimental testing? Why not combined with/only computational simulation and/or clinical study?
Answer: Our work is about the first attempt to choose the perspective Mg-Zn-Ga alloys and its processing parameters. In prospective, after corrosion properties and biocompatibility investigation the Young modulus and Poisson ratio of perspective alloy will be found and deep analysis of the mechanical behavior of the biodegradable tools made of this alloys by simulation via finite element method will be done.
- Since the present study only perform experimental testing, the potential of computational simulation study via finite element method is needed to discuss since bring some advantages such as faster results and do not need sophisticate tools. The introduction and/or discussion part of an article should contain this crucial topic, according to the authors. In addition, to reinforce this explanation, the MDPI-recommended reference should be cite as follows: Ammarullah, M. I.; Afif, I. Y.; Maula, M. I.; Winarni, T. I.; Tauviqirrahman, M.; Jamari, J. Tresca Stress Evaluation of Metal-on-UHMWPE Total Hip Arthroplasty during Peak Loading from Normal Walking Activity. Mater. Today Proc. 2022, 63, S143–6. https://doi.org/10.1016/j.matpr.2022.02.055
Answer: The Ammarullah et al work is about investigation of mechanical behavior of fully constructed tool form materials with well-known properties. Maybe in our future works this reference will be actual, but not at initial stage of our investigation where the material is chosen and the geometry of biodegradable tools are unknown.
- In order to improve the reader's understanding of the materials and methods section simpler, the authors could provide a figures that clarify the workflow of the current study rather than only the predominant text as it currently appears.
Answer: The scheme of extruded bars processing from casting to hot extrusion was added to the materials and methods section.
- Additional information about tools used, such as the maker, country, and specification, should be included.
Answer: The maker and country was added where missed.
- The error and tolerance of the experimental tools used in this investigation are important aspects that have to be mentioned in the manuscript. It might be valuable for further research by other scholars because of the different results.
Answer: We add the experimental tool tolerance where it is possible (for EDS and mechanical test). In most cases the error bar is higher than the tool error.
- Outcomes must be compared to similar past research.
Answer: Outcomes compared with previous research in the end of Results and discussion part.
- Before moving on to the conclusion section, the present study's limitation must be added at end of the discussion section.
Answer: The limitations was added: “This study was the first attempt to investigate the influence of hot extrusion on the microstructure and mechanical properties of newly developed Mg–Zn–Ga–(Y) alloys. For better understanding of the mechanical behavior of the mentioned alloys the extensive texture analysis and Visco-Plastic Self-Consistent (VPSC) simulations are needed. Fur-thermore the implants work in the human body and the mechanical properties in the body fluids corrosive environment such as stress corrosion cracking and corrosion fatigue must be known for Mg–Zn–Ga–(Y) alloys in order to use them as biodegradable materials.”.
- In the conclusion, please explain the further research.
Answer: This information was added: “The further research will focus on in vitro corrosion behavior and biocompatibility of Mg–Zn–Ga–(Y) alloys to choose the best composition. After that the coating must be developed for reduction in biodegradation rate and gas formation, that provide a better bone integration. Finally, the in vivo test on animals to show the increased bone growth rate and improved healing process in comparison with other magnesium alloys will be con-ducted.”
- The reference should be enriched with literature from the last five years. Literature published by MDPI is strongly recommended.
Answers: The references 2020-2022 to MDPI articles was added to the manuscript.
- Due to grammatical and linguistic style issues, the authors should proofread the manuscript. For this issue, the authors would utilize the MDPI English editing service.
Answer: The manuscript was proofread.
- Please ensure that the writers followed the MDPI format correctly, modify the current form, and recheck in addition to any other problems that have been identified.
Answer: It was fixed.

Reviewer 2 Report
Dear authors, I consider that your manuscript needs major revision. Please see the remarks presented in the attached review document.

Author Response
Review on manuscript materials-1916966
of paper “Microstructure and mechanical properties of hot-extruded Mg-Zn-Ga-(Y) biodegradable
alloys”
The article studies some properties of hot-extruded magnesium alloys used for medical applications in temporary biodegradable implants and the theme it is in accordance with journal topics.
The title reflects the article content and the abstract briefly summarizes the purpose of the research. After the introduction on the state of the art on studies of different magnesium alloys, the authors present subsequently the materials and method used in this research, results and extensive discussions on the quality of extruded samples, microstructure, phase composition and mechanical properties. The complex analysis and experimental investigations presented in this manuscript shows the efforts made by the authors.
However, some statements are unclear and the manuscript needs revision and improvement.
Remark 1:
The introduction section must be improved. A comprehensive and exhaustive review of the state of the art in the field of the study must be provided. Please introduce and discuss more recent works, and highlight the experiments and results published previously. Also, in the last paragraph of this section please highlight the novelty and importance in science advance of this study.
Answer: The introduction was rewritten.
Remark 2:
Keywords must indicate the main materials, tests, and methodology used in the study. However, it is required to revise the keywords and write based on the points mentioned above.
Answer: The keywords was changed to: biomaterials; gallium; hot extrusion; magnesium; mechanical properties; microstructure
Remark 3:
In figure 9 what represent engineering stress and strain??? These two parameters are obtained for a tensile test of the extruded magnesium alloys samples and I think that represent the tensile stress and strain!!!
This confusing also the readers!!!
Answer: We add “tensile” to the figure and figure caption.
Figure 9. Engineering tensile stress–strain curves for the Mg–Zn–Ga–(Y) alloys extruded at 200 or 150 °C.
Remark 4:
In figure 10 replace on y axis the acronyms with text “Tensile tensile strength”, “Ultimate tensile strength”, “Elongation”, etc. (for reader’s better understanding)! !!!
Answer: We agree with this comment and replaced acronyms with text on “y” axis on Fig.10.
Remark 5:
The conclusions are very general and need to be revised and improved. Please make sure the conclusion section underscores the scientific value added by the article and the applicability of the findings/results.
Answer: The conclusion was rewritten.
In conclusion, I consider that the article have some errors and present a medium to low level of novelty.
Due to remarks presented above, I recommend major revision of the manuscript.

Reviewer 3 Report
1. Is the high-purity bulk metal bought from some company or prepared by the author's group? If prepared in the lab, can the author provide some characterization for it's high purity? If bought, can the author provide which company to ensure the repeatability ?
2. Can the author explains more on why the three extrusion temperature is selected for this study? (150, 200, 250)
3. Abstract is too rough. Can the author polish more on the abstract? The author mentions Mg-Zn-Ga-Y alloy yields favorable properties. The author should summarize it in the abstract and stress the key advantage. Overall, the abstract is poorly written.
4. Background should clearly indicate why the Mg alloy is better than Ti Alloy. First, Ti alloy foam can also process similar density to the bone, secondly, Ti alloy is very corrosion resistant. Can the author provide more background related to the Mg alloy and clearly identifies the key advantage?
5. For the reference part, most citations are old. Can the author include more on current research in this field?
Author Response
- Is the high-purity bulk metal bought from some company or prepared by the author's group? If prepared in the lab, can the author provide some characterization for it's high purity? If bought, can the author provide which company to ensure the repeatability?
Answer: The pure material was bought, information is provided.
- Can the author explains more on why the three extrusion temperature is selected for this study? (150, 200, 250)
Answer: This information was added to the introduction part of the manuscript: “The extrusion processing temperature window determined by the possibilities of used equipment (possible lowest extrusion temperature) and limit of hot cracking (possible highest extrusion temperature) [49]. For Mg–Zn–Ga–(Y) alloys the solidus temperature close to 300 °C, that was shown previously [24] and confirmed via CALPHAD calculation and DSC in this work. Because of that the upper limit of extrusion temperature 250 °C was chosen. The lower limit of 150 °C was chosen in accordance with the maximal pressure of the used press.”
- Abstract is too rough. Can the author polish more on the abstract? The author mentions Mg-Zn-Ga-Y alloy yields favorable properties. The author should summarize it in the abstract and stress the key advantage. Overall, the abstract is poorly written.
Answer: Abstract was rewritten.
- Background should clearly indicate why the Mg alloy is better than Ti Alloy. First, Ti alloy foam can also process similar density to the bone, secondly, Ti alloy is very corrosion resistant. Can the author provide more background related to the Mg alloy and clearly identifies the key advantage?
Answer: This information was added to the introduction: “Permanent bone fixation implants are gold standard in osteosynthesis and used in healthcare systems in many countries. However, the titanium systems have disadvantages include temperature sensitivity, tactile sensation of implants, possible growth restrictions, hampering of imaging and radiotherapy, presence of titanium particles in surrounding tissue, and potential mutagenicity [49]. These disadvantages result in symptomatic removal in up to 40% of cases [49].”
- For the reference part, most citations are old. Can the author include more on current research in this field?
Answer: The references 2020-2022 was added to the manuscript.

Round 2
Reviewer 1 Report
Reviewers greatly appreciate the efforts that have been made by the author to improve the quality of their articles after peer review. I reread the author's manuscript and further reviewed the changes made along with the responses from previous reviewers' comments. Unfortunately, the authors failed to make some of the substantial improvements they should have made making this article not of decent quality with biased, not cutting-edge updates on the research topic outlined. In addition, the author also failed to address the previous reviewer's comments, especially on comments number 8, 9, 10, and 11. With all due respect, the reviewer opposed this article to be published and must be rejected. Thank you very much for the opportunity to read the author's current work.
Author Response
Comments made by Reviewer#1 on the round 2
Comments and Suggestions for Authors
Reviewers greatly appreciate the efforts that have been made by the author to improve the quality of their articles after peer review. I reread the author's manuscript and further reviewed the changes made along with the responses from previous reviewers' comments. Unfortunately, the authors failed to make some of the substantial improvements they should have made making this article not of decent quality with biased, not cutting-edge updates on the research topic outlined. In addition, the author also failed to address the previous reviewer's comments, especially on comments number 8, 9, 10, and 11. With all due respect, the reviewer opposed this article to be published and must be rejected. Thank you very much for the opportunity to read the author's current work.
Answer: We try to answer to 8-11 comments again and add some changes to the manuscript.
- The novel of the present study is not clear. Many published literature has been widely studied in the past related to magnesium based biodegradable alloys. Further explanation in the introduction section in advance is mandatory.
Answer: The introduction was rewritten. It was shown that Mg–Zn–Ga–(Y) alloys in bone fixation implants potentially can improve the bone healing process due to presence of Ga that is the benefit in comparison with other conventional magnesium alloys. Also in introduction we indicate that this work is the first systematic study on microstructure and mechanical properties of Mg-Zn-(Y) alloys containing Ga as an additional major alloying element prepared by the hot-extrusion process. The hot extrusion processing is chosen due to advantages compared with ECAP processing, such as less limitations in size and shape on the billet, easy control of microstructure, low processing cost, etc.
- Previous research has to be explained in the introduction section, including their work, novelty, and limits, to illustrate the research gaps that will be filled in the current study.
Answer: The information about the commercial NOVAMag® and MAGNEZIX® fixation screws produced by Botiss biomaterials GmbH and Syntellix AG (both Germany) is provided that show the actual state-of-the-art level. Also the data about the good influence of Ga on magnesium alloys (microstructure, mechanical and corrosion properties) are provided. The information about the Ga effect on the bone growth showing the potential of investigated alloys is also present.
- Why the present study perform experimental testing? Why not combined with/only computational simulation and/or clinical study?
Answer: This work is the first attempt to examine the effects of various processing parameters (composition, extrusion temperature, speed, heat treatment condition, etc.) on microstructure and mechanical properties of newly proposed biodegradable Mg-Zn-Ga-(Y) alloys, which could not be predicted by computational simulation at present. In prospective, after corrosion properties and biocompatibility investigation the Young modulus and Poisson ratio of perspective alloy will be found and deep analysis of the mechanical behavior of the biodegradable tools made of this alloys by simulation via finite element method will be done.
- Since the present study only perform experimental testing, the potential of computational simulation study via finite element method is needed to discuss since bring some advantages such as faster results and do not need sophisticate tools. The introduction and/or discussion part of an article should contain this crucial topic, according to the authors. In addition, to reinforce this explanation, the MDPI-recommended reference should be cite as follows: Ammarullah, M. I.; Afif, I. Y.; Maula, M. I.; Winarni, T. I.; Tauviqirrahman, M.; Jamari, J. Tresca Stress Evaluation of Metal-on-UHMWPE Total Hip Arthroplasty during Peak Loading from Normal Walking Activity. Mater. Today Proc. 2022, 63, S143–6. https://doi.org/10.1016/j.matpr.2022.02.055
Answer: The Ammarullah et al work is about investigation of mechanical behavior of fully constructed tool form materials with well-known properties. This work is the first attempt to examine the effects of various processing parameters (composition, extrusion temperature, speed, heat treatment condition, etc.) on microstructure and mechanical properties of newly proposed biodegradable Mg-Zn-Ga-(Y) alloys, which could not be predicted by computational simulation at present. Maybe in our future works this reference will be actual, but not at initial stage of our investigation where the material is chosen and the geometry of biodegradable tools are unknown.
Reviewer 2 Report
Dear authors, I see major improvements of your manuscript. You answered clearly and satisfactory to all my review remarks. I recommend publishing the manuscript in the MDPI journal.
Author Response
Comments and Suggestions for Authors
Dear authors, I see major improvements of your manuscript. You answered clearly and satisfactory to all my review remarks. I recommend publishing the manuscript in the MDPI journal.
Answer: We want to thank the reviewer for important suggestions.
Round 3
Reviewer 1 Report
The present study still lacks novelty and scientific contribution. Nothing something new was established with the present paper. The discussion is also not in-depth with cutting-edge insight. The Reviewer is against this manuscript for publication and must be rejected.